# Technical Note: False low turbidity readings from optical probes during high suspended-sediment concentrations

Nicholas Voichick[1], David J. Topping[1], and Ronald E. Griffiths[1]

[1]U.S. Geological Survey, Grand Canyon Monitoring and Research Center, 2255 N. Gemini Dr., Flagstaff, Arizona, 86001, USA

*Correspondence to*: Nicholas Voichick (nvoichick@usgs.gov)

**Abstract.** Turbidity, a measure of water clarity, is monitored for a variety of purposes including: 1) to help determine whether water is safe to drink; 2) to establish background conditions of lakes and rivers and detect pollution caused by construction projects and stormwater discharge; 3) to study sediment transport in rivers and erosion in catchments; 4) to manage siltation of water reservoirs; and 5) to establish connections with aquatic biological properties, such as primary production and predator-prey interactions. Turbidity is typically measured with an optical probe that detects light scattered from particles in the water. Probes have defined upper limits of the range of turbidity that they can measure. The general assumption is that when turbidity exceeds this upper limit, the values of turbidity will be constant, i.e., the probe is "pegged"; however, this assumption is not necessarily valid. In rivers with limited variation in the physical properties of the suspended sediment, at lower suspended-sediment concentrations, an increase in suspended-sediment concentration will cause a linear increase in turbidity. When the suspended-sediment concentration in these rivers is high, turbidity levels can exceed the upper measurement limit of an optical probe and record a constant "pegged" value. However, at extremely high suspended-sediment concentrations, optical turbidity probes do not necessarily stay "pegged" at a constant value. Data from the Colorado River in Grand Canyon, Arizona, USA and a laboratory experiment both demonstrate that when turbidity exceeds instrument-pegged conditions, increasing suspended-sediment concentration (and thus increasing turbidity) may cause optical probes to record decreasing "false" turbidity values that appear to be within the valid measurement range of the probe. Therefore, under high-turbidity conditions, other surrogate measurements of turbidity (e.g., acoustic-attenuation measurements or suspended-sediment samples) are necessary to correct these low false turbidity measurements and accurately measure turbidity.

## 1 Introduction

Turbidity, a measure of the scattering and absorption of light in water, is dependent on the characteristics of the particles that are scattering the light, specifically concentration, grain size, grain shape, refractive index, and color (Sadar, 1998; Downing, 2006). For example, clay-sized particles result in much greater turbidity than an equal concentration of sand-size particles (Davies-Colley et al., 1993; Gippel, 1989). Although turbidity is commonly used to monitor change in water clarity, it is not an absolute measure of water clarity because different models of turbidity instruments can give different readings in the same

water. The design components of a turbidity instrument that affect the range of measurements are the light source, the detector(s), the optical geometry, which includes the path length of the light, and the angle of the detector(s) from the incident light path (Hach et al., 1985; ASTM International, 2011).

A commonly used turbidity probe on multi-parameter water-quality instruments is a nephelometric turbidity probe, which measures light scattering. A typical configuration is a probe with a single light source in the near infra-red wavelength range and single detector oriented 90 degrees from the incident light path (Fig. 1). Along with turbidity, additional parameters frequently measured from the same water-quality instrument include temperature, specific conductance, and dissolved oxygen. These water-quality instruments are particularly useful for detecting variation from typical conditions in rivers and lakes. For

example, such instruments have been used for drinking water protection by being deployed upstream from water treatment plants to monitor for high-turbidity runoff events during storms (U.S. Environmental Protection Agency, 2016). The instruments can be set up to trigger alarms when turbidity or other water-quality parameters exceed permitted thresholds, allowing operators to shut down drinking water intakes to protect filtration systems. These water-quality instruments have also been used to monitor and regulate turbidity caused by construction and dredging projects in and around natural bodies of water.

Turbidity data in the Colorado River are primarily used by biologists to relate to primary production and interactions between native endangered fish and predatory non-native fish competing for limited resources (Hall et al., 2015; Wellard Kelley et al., 2013; Ward et al., 2016; Yard et al., 2011).

With a constant composition and grain-size distribution of suspended sediment, the turbidity response of a particular instrument

varies depending on the suspended-sediment concentration. Figure 2 shows a generic response curve of a single-detector instrument in turbidity units; the specific measurement unit depends on the properties of the instrument (such as wavelength of incident light and detector angle; ASTM International, 2011). At low concentrations (or low "Percent of Original Sample", Fig. 2), turbidity measured from a single-detector instrument increases linearly with increasing concentration. At high concentrations, turbidity plateaus at the maximum recording level when the detector becomes saturated with light. In some

cases, at still higher concentrations, the recorded turbidity may decrease as light penetration into the sample diminishes (i.e. a negative response, Fig. 2; ASTM International, 2011). This negative response (i.e., incorrectly low turbidity) is characterized by a high percentage of the light being absorbed by the suspended sediment (or other material) and less light reaching the detector, resulting in progressively lower turbidity readings with increasing concentration. These incorrectly low turbidity readings at high concentrations will hereafter be referred to as false low turbidity. Turbidity measured in the Colorado River

in Grand Canyon, Arizona, USA occasionally records as false low turbidity at very high suspended-sediment concentrations (Fig. 5). Without other measures of turbidity, these false low turbidity readings can be confused with similar correct turbidity values recorded during the linear portion of Figure 2 at much lower suspended-sediment concentration.

This study describes both field and laboratory conditions that lead to false low turbidity readings. Recognition of false low turbidity is important because the high-turbidity conditions that give rise to such measurements could inadvertently result in environmental harm as well as expensive equipment damage. For example, when turbidity is being monitored on a river below a construction site, false low turbidity would underrepresent a high sediment load which may be considered an environmental contaminant. Additionally, the undetected high sediment load could damage, for example, a drinking water filtration system.

## 2 Methods

### 2.1 Field sites

Six gaging stations have been established in the study area in the Colorado River in Grand Canyon National Park, Arizona, USA (Fig. 3) to monitor suspended-sediment load as well as water-quality parameters, specifically water temperature, specific conductance, dissolved oxygen, and turbidity. These water-quality parameters are measured using YSI Incorporated (YSI) multi-parameter sondes with turbidity measured using YSI 6136 turbidity probes (the use of firm, trade, and brand names is for identification purposes only and does not constitute endorsement by the USGS). Changes in turbidity of the Colorado River in the study area are primarily caused by changes in suspended-silt-and-clay concentration (Voichick and Topping, 2014). Owing to the large range in silt and clay concentration possible in this river, optical turbidity probes exhibit pegged and false low turbidity on many days per year. Acoustic attenuation from side-looking acoustic-Doppler profilers is used as a surrogate measure of suspended-silt-and-clay concentration and turbidity in the study area under such pegged and false low turbidity conditions because this method has been shown to provide accurate measurements of silt and clay concentrations (Topping et al., 2007; Voichick and Topping, 2014; Topping and Wright, 2016). Acoustical data and data from the water-quality instruments are collected concurrently every 15 minutes at all stations in the study area. Physical suspended-sediment samples are collected episodically (using Equal-Width-Increment, Equal-Discharge-Increment, and automatic-pump methods described in Edwards and Glysson (1999)) to verify the acoustical and turbidity measurements.

### 2.2 Laboratory experiment

The largest sources of silt and clay in the study area are the Little Colorado River and the Paria River (Topping et al., 2000, Voichick and Topping, 2014). The sediment used to create high-turbidity conditions in the laboratory was collected from the bank of the Little Colorado River approximately 1 kilometer upstream from its confluence with the Colorado River (Fig. 3). The sediment was passed through a 63-µm sieve to isolate the silt-and-clay-sized fraction. Only silt-and-clay-sized sediment was used in the experiment because: 1) silt and clay causes almost all of the turbidity in the study area (Voichick and Topping, 2014), and 2) by eliminating sand, it was much easier to keep the sediment in suspension to achieve accurate turbidity measurements. In addition, silt-and-clay concentration was measured at the gaging stations from suspended-sediment samples and from acoustical measurements (concurrently with turbidity), allowing for comparison of laboratory and field results. The silt and clay was completely dried in an oven (at 105 degrees Celsius for 12 hours), weighed, mechanically disaggregated, and

added in stages to a measured volume of water to calculate concentration. The sediment was kept in suspension in a 20-liter bucket using an electric stirrer (the Agitator, manufactured by Arrow Engineering) with three 2.5-cm-long blades placed approximately 1 cm above the bottom of the bucket. The water-quality instrument and turbidity probe used in the laboratory experiment were the same models as were present in the study area, a YSI 6920 instrument and 6136 turbidity probe. With the

probe guard attached, the instrument was suspended in the bucket with the turbidity probe located approximately 9 cm above the bottom of the bucket. The speed of the stirrer was increased until turbidity readings were stable. Sediment concentration, in mg/L, and the corresponding turbidity value in Formazin Nephelometeric Units (FNU) were recorded.

## 3 Results

### 3.1 Field data

Between storm-driven tributary floods, the Colorado River in the study area is relatively clear, with turbidity less than 15 FNU for 50 percent of the time (averaged over all stations, Voichick and Topping, 2014). Turbidity was elevated to the probe's maximum recording level (Figs. 2 and 4) from less than 1 percent of the time at the furthest upstream station (Colorado River at Lees Ferry, Arizona) to 7 percent of the time at the furthest downstream station (Colorado River above Diamond Creek near Peach Springs, Arizona) over a period of 5 to 8 years depending on the station (Voichick and Topping, 2014). False low

turbidity was recorded during 5 percent of the tributary flooding events when actual turbidity was above the probe's maximum recording level (Figs. 2 and 4). Consistent with the laboratory experiment, false low turbidity was recorded in all cases during the highest sediment concentrations of a flooding event, with the lowest false low turbidity recorded at the peak sediment concentration (Fig. 5). The lowest silt-and-clay concentrations during the false low turbidity readings ranged from 17,000 mg/L to 27,000 mg/L, averaging 21,200 mg/L (Fig. 5). However, not all flooding events with high silt-and-clay concentration

resulted in false low turbidity; 50 percent of flooding events with silt-and-clay concentrations greater than 20,000 mg/L showed no false low turbidity (i.e., turbidity remained "pegged" at the maximum recording level in these cases).

### 3.2 Laboratory experiment

At lower silt-and-clay concentration, the laboratory experiment showed a linear increase in turbidity with increasing silt-and-clay concentration (Fig. 4). Starting at a silt-and-clay concentration of ~3,000 mg/L, turbidity pegged and recorded at the

maximum recording level until false low turbidity was initially recorded at a silt-and-clay concentration of ~38,000 mg/L, above which measured turbidity incorrectly decreased nonlinearly with increasing silt-and-clay concentration.

## 4 Discussion and conclusions

Data from the Colorado River in the Grand Canyon study area agree with laboratory results showing that when deploying a turbidity probe with a single 90-degree oriented detector, false low turbidity is possible at high suspended-sediment

concentrations (Figs. 2, 4, and 5). The working range of a turbidity probe is the linear portion of the response curve, when turbidity increases linearly with increasing sediment concentration (assuming constant sediment characteristics, Figs. 2 and 4). This is generally stated as a particular range of turbidity values. For the turbidity probe used in the Grand Canyon study area, the working range defined by the manufacturer is 0 to 1,000 FNU. Importantly, the laboratory experiment and field data from the Grand Canyon study area showed that many of the false low turbidity readings were reporting within the apparent working range of the turbidity probe. Thus, these readings could unknowingly be interpreted as correct (and relatively low) turbidity even though they represent some of the highest suspended-sediment concentrations (and thus some of the highest values of turbidity) observed at the particular site, substantially higher than the working range of the turbidity probe. In the Grand Canyon study area, acoustic attenuation is used to estimate turbidity above the working range of the probe, based on a linear relation between turbidity and acoustic attenuation (Voichick and Topping, 2014). In the example shown in Figure 5, the actual turbidity based on concurrent acoustical data was up to 60 times greater than the probe's false low turbidity readings.

Whereas turbidity readings at the maximum recording level of the probe are easily recognized, false low turbidity could be confused with similar turbidity recorded at much lower sediment concentrations. The sediment concentration at which false low turbidity occurs will depend on the characteristics of the sediment as well as the properties of the instrument. In the Grand Canyon study area, the sediment characteristics which likely have the greatest effect on turbidity measured with the YSI instruments are concentration and grain-size, with finer grains resulting in higher turbidity (and perhaps a lower sediment concentration for false low turbidity readings) than an equal concentration of coarser grains (Voichick and Topping, 2014). In the Grand Canyon study area, the silt-and-clay concentration threshold values when false low turbidity occurred ranged from 17,000 to 27,000 mg/L, whereas in the laboratory experiment, false low turbidity was initially recorded at approximately 38,000 mg/L silt-and-clay concentration. The most likely explanation for the difference in threshold concentration for false low turbidity between the field and laboratory is a difference in the grain-size distribution of the silt and clay. Most clay-size sediment in suspension in the field area is transported as washload and never gets deposited. Thus, it is highly likely that the bank-collected sample used in the laboratory experiment was coarser and composed of a higher percentage of silt-size sediment than is typically present in suspension in the field area. Regardless of the exact cause of the difference in threshold concentration between the field and laboratory results, there was a large range of threshold silt-and-clay concentrations in the study area when false low turbidity initially occurred (from 17,000 to 27,000 mg/L); this was likely because of variations in the silt-and-clay grain-size distribution as well as variation in other sediment characteristics, perhaps the result of differing sediment sources (Voichick and Topping, 2014).

This study demonstrates false low turbidity recorded in the Grand Canyon study area and in the laboratory using a single-detector turbidity probe with a near infra-red wavelength light source, measuring light scattered at 90 degrees from the incident light path. It is possible that other types of single-detector turbidity probes, with a different wavelength light source and/or detector angle, could also record false low turbidity. If a particular instrument "pegs" at high sediment concentrations when

the detector is saturated with light, it is conceivable that when sediment concentration is further increased, light received by the detector could be reduced and result in false low turbidity (as happens with the turbidity probes used in this study). Although selecting a turbidity probe with multiple detectors could eliminate the possibility of false low turbidity, because the detectors receiving light at different scattering angles would react differently to highly concentrated samples, multiple detector probes are not always a practical choice. For example, the single-detector turbidity probe used in the Grand Canyon study area is the only probe compatible with the YSI water-quality instrument used in the study area.

In some rivers, including the Colorado River in Grand Canyon, suspended-sediment concentration is not well correlated with discharge because most or all of the suspended sediment is contributed from tributaries that contribute a large amount of sediment but little discharge to the mainstem river. In these rivers, discharge cannot be used to estimate suspended-sediment concentration. Thus, false low turbidity is often difficult to recognize without an alternative method of measuring turbidity, such as acoustically or by collecting and analysing physical suspended-sediment samples. The consequences of not recognizing false low turbidity are: 1) underestimating turbidity, and 2) missing the timing of the peak sediment concentration of a flooding event. Additionally, because false low turbidity occurs only during the highest suspended-sediment concentrations, false low turbidity occurs during floods with the largest sediment loads. In the example shown in Figure 5, false low turbidity occurred during the period when 70 percent of the suspended-sediment load during this flooding event passed the gaging station. Without additional evidence, a period of false low turbidity (Fig. 5) would likely be incorrectly interpreted as a period of lower turbidity between two flooding events instead of the period of actual maximum turbidity and suspended-sediment concentration during a single large flooding event. Without knowing that false low turbidity does not represent the actual turbidity, water could, for example, mistakenly be drawn into a water-treatment plant and cause damage to the filtration system. If associated with the monitoring of a construction or dredging project, false low turbidity could result in regulators being unaware of environmental damage caused by the actual, much-higher turbidity. In cases where turbidity is monitored to link water clarity with biological processes such as primary production or fish behavior, false low turbidity could result in an incorrect interpretation of these data relations.

Our results suggest that when turbidity readings within the valid measurement range of the probe are bracketed by pegged turbidity readings, false low turbidity should be suspected. This pattern would be expected because false low turbidity only occurs at the highest suspended-sediment concentrations and would thus most likely be preceded by and followed by pegged turbidity. Data showing this pattern should be verified using surrogate measures of turbidity, such as acoustic attenuation or suspended-sediment concentration, especially if suspended-sediment concentration is known or suspected to be particularly high (e.g. greater than several thousand mg/L).

**5 Competing interests**

The authors declare that they have no conflict of interest.

**6 Data availability**

Gaging-station data can be accessed, plotted, and downloaded at: https://www.gcmrc.gov/discharge_qw_sediment/ or https://cida.usgs.gov/gcmrc/discharge_qw_sediment/. Data associated with the suspended-sediment laboratory experiment are available from the USGS ScienceBase-Catalog at: https://doi.org/10.5066/F72N516S.

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

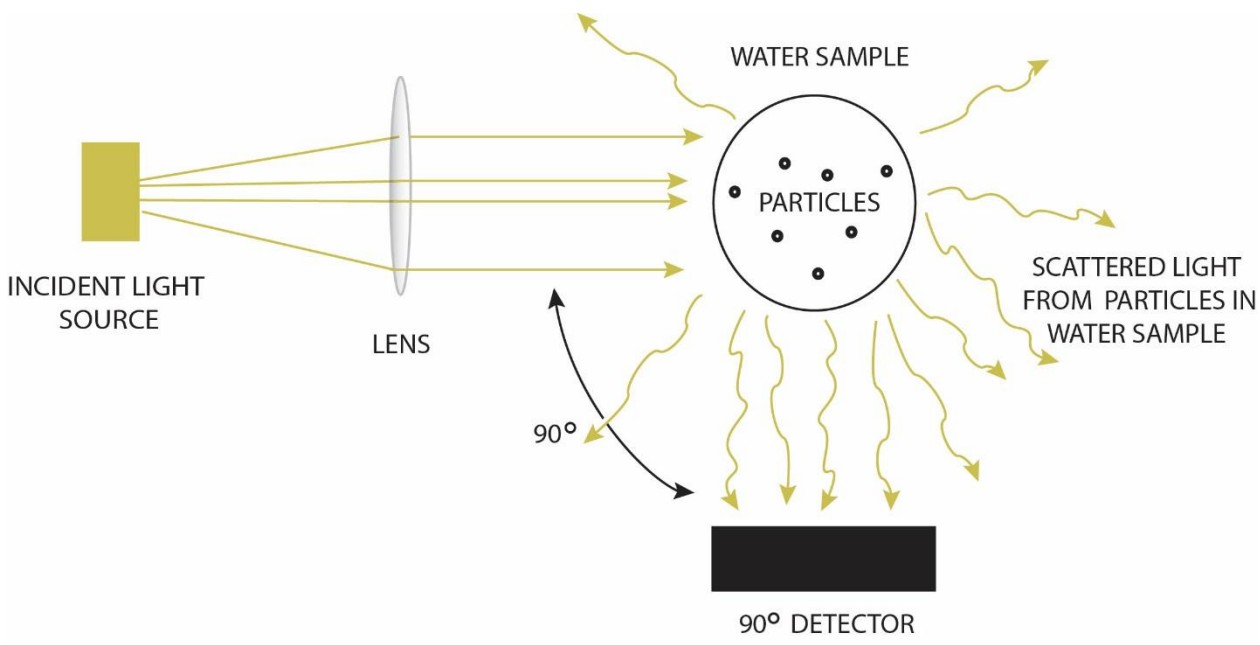

**Figure 1: Diagram showing the design components of a nephelometric turbidity instrument: the light source, the detector, and the optical geometry. The diagram shows a single detector oriented 90 degrees from the incident light path, which was used in the Colorado River study described in this paper (modified from Sadar, 2009).**

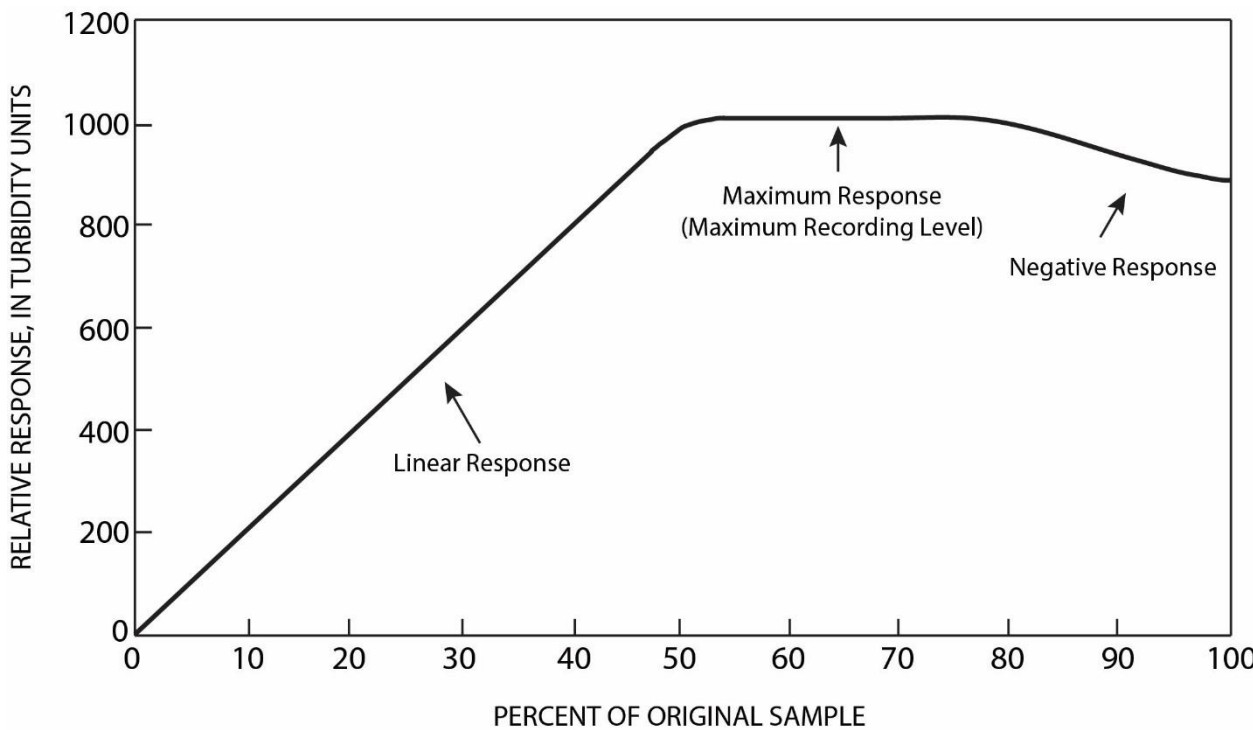

**Figure 2: Generic response curve of a single-detector turbidity instrument showing three turbidity responses. The turbidity response is dependent on the characteristics of the turbidity instrument as well as the concentration of material (e.g., sediment) in the sample. "Original Sample" on the horizontal axis refers to a water sample containing a high concentration of sediment or some other light-scattering material. As one moves from right to left on this axis, this original sample is diluted with increasing amounts of clear turbidity-free water. "0" thus indicates clear water and increasing numbers indicate increasing sediment or particle concentrations (modified from ASTM, 2011).**

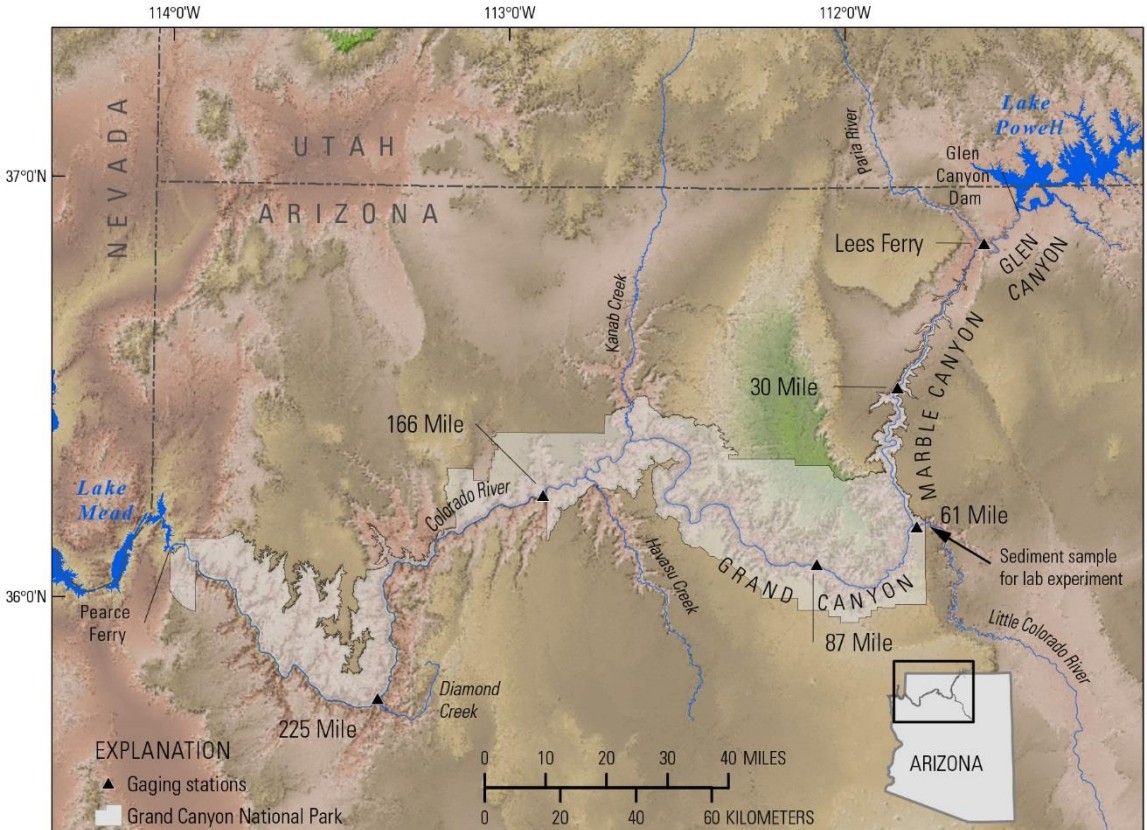

**Figure 3: Map of northwestern Arizona, showing the study area in Grand Canyon, major tributaries, and the location of the gaging stations. Gaging stations at which turbidity, suspended sediment and other water-quality data were collected are: Colorado River at Lees Ferry, AZ, 09380000 (abbreviated as Lees Ferry); Colorado River near river mile 30, 09383050 (abbreviated as 30 Mile); Colorado River above Little Colorado River near Desert View, AZ, 09383100 (abbreviated as 61 Mile); Colorado River near Grand Canyon, AZ, 09402500 (abbreviated as 87 Mile); Colorado River above National Canyon near Supai, AZ, 09404120 (abbreviated as 166 Mile); and Colorado River above Diamond Creek near Peach Springs, AZ, 09404200 (abbreviated as 225 Mile).**

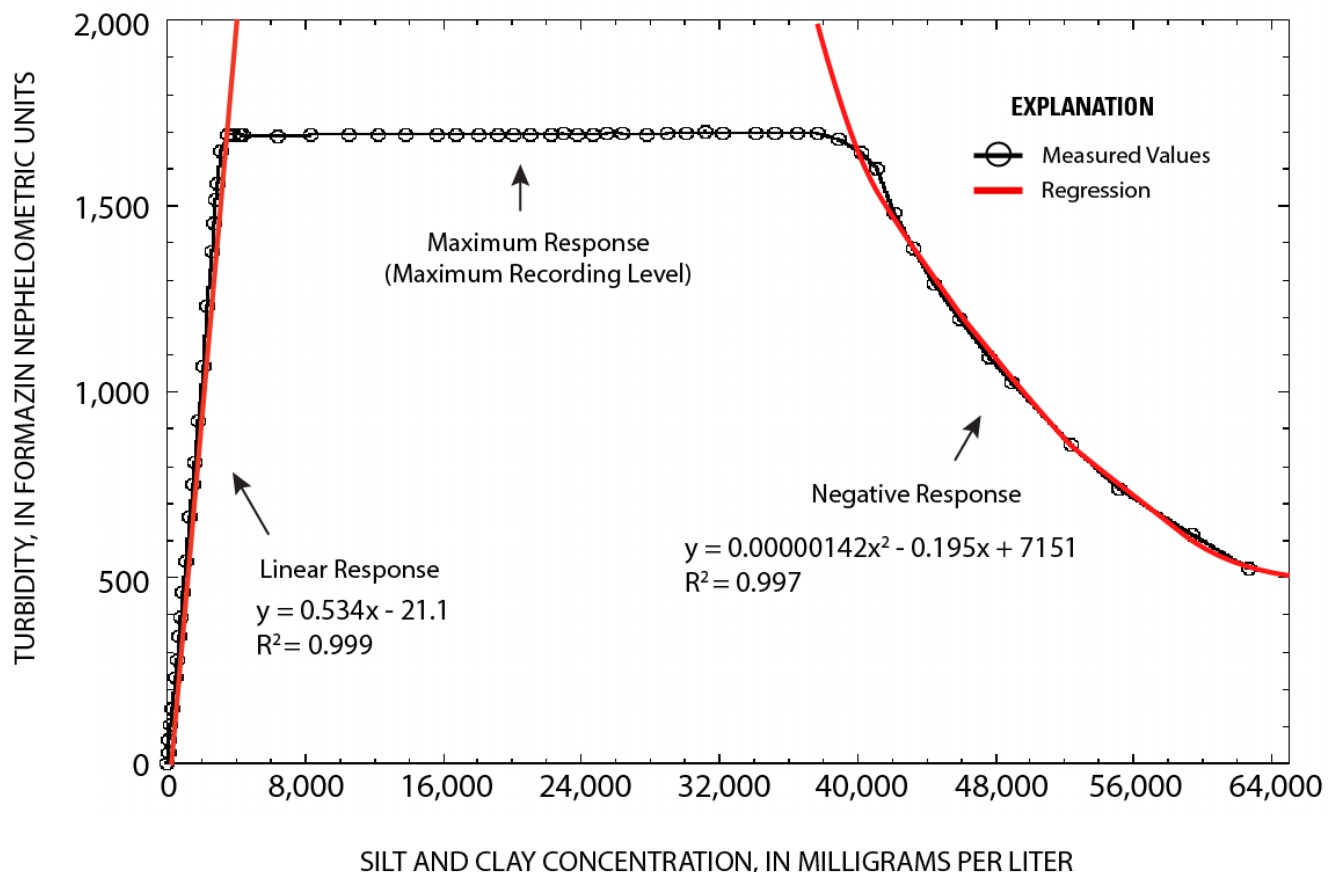

**Figure 4: Graph showing the relation between silt-and-clay concentration and turbidity during a laboratory experiment. The silt and clay was obtained from the bank of the Little Colorado River (Fig. 3) and turbidity was measured using a YSI Incorporated model 6136 probe. The fitted curves for the linear and negative response portions of the graph are shown in red.**

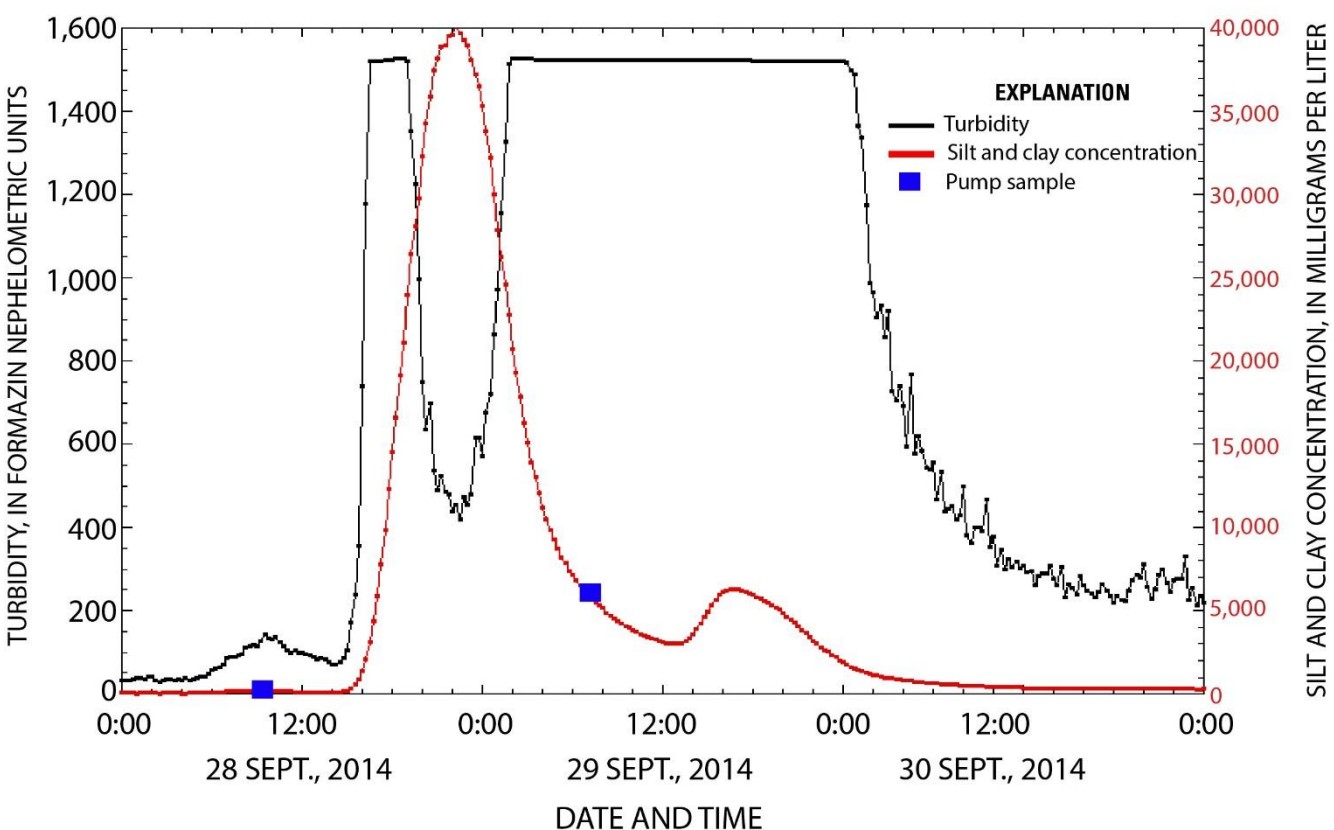

**Figure 5: Graph of turbidity (shown in black) and silt-and-clay concentration measured with a 1 MHz acoustic-Doppler instrument (shown in red, pump sample silt and clay concentration shown in blue) at the Colorado River near river mile 30 gaging station (Fig. 3) from September 28 to October 1, 2014. Turbidity changed from a linear response to increasing silt-and-clay concentration, to a maximum response (i.e. pegged readings), to a negative turbidity response (defined as false low turbidity), which first occurred at 19:15 on September 28 when the concentration was 26,400 mg/L. The false low turbidity continued as silt-and-clay concentration increased to a peak of 40,100 mg/L at 22:15 on September 28, and then decreased to 24,600 mg/L at 1:30 on September 29. As this suspended-sediment pulse travelled downstream, a similar false low turbidity was seen at all four downstream gaging stations (Fig. 3).**