# Peer review of "Technical Note: False low turbidity readings from optical probes during high suspended-sediment concentrations"

_Hydrology and Earth System Sciences, 2017_

## Referee Comment (RC1) · Anonymous Referee #1 · 21 Nov 2017

The present manuscript contains a study on the limitations and, most importantly, on the false results that one can obtain using an optical turbidimeter. Results from field measurements and from a laboratory investigation are presented and discussed. The authors use other surrogates of turbidity to compare with the values obtained with an optical sensor. It is surely an extremely useful topic and within the scope of HESS.

The manuscript is thus a valid and neat contribution to the community and it reads as a technical report, the category of submission. The manuscript is generally well written and well organized. I recommend acceptance of the manuscript after solving a few points for improvement and discussion which I present next.

In the title reference to optical probes should be given. I think this is the main focus of the paper, it should be emphasized.

In the abstract, although said in line 11, the authors should somehow stress that their analysis is related with optical probes, only later in line 24 this is clear.

In lines 16-20 it is a bit confusing here the reference to the physical properties of the sediment and later the relation with the fact that some devices do not peg. Rephrase these two sentences.

Lines 32 and 33, the sentence somehow seems contradictory.

In line 104, more details should be given on the physical suspended-sediment samples should be given. Which method was used for these other surrogate measurements?

In figure 4 there is no legend regarding the information presented. What are all the lines and symbols herein represented?

The discussion regarding the dependence of the sediment concentration measurements and the characteristics of sediment in line 174-175 should be more complete. In what sense these are related to the properties of the instrument? This is a crucial point since this may actually lead to improvement of the techniques.

The suggestion given in lines 200-201 is not clear, needs to be improved and more complete.

I suggest to the authors the reading of Gitto et al. (2017), it may give them some useful complementary information.

References: Gitto, A. B., Venditti, J. G., Kostaschuk, R., & Church, M. (2017). Representative point‐integrated suspended sediment sampling in rivers. Water Resources Research, 53(4), 2956-2971.

528, 2017.

---

## Author Comment (AC1) · 22 Nov 2017

Thank you for your useful comments. I think that your comments and suggestions will improve the paper. However, I am not sure what part of the manuscript all of the comments refer to because when I download the manuscript, the line numbers do not match up with the line numbers you refer to. For example, the downloaded manuscript only has 155 lines to the start of the Reference section but you refer to "lines 174-175 and 200-201. Please include some of the sentence that you are referring to with the line numbers, in particular:

1) What sentence is being referred to in "Lines 32 and 33"?

[Figure]

2) What sentence is "line 104" referring to?

3) I think that "line 174-175" refers to the sentence starting out with "The sediment concentration at which false low turbidity....". Please verify this.

4) What sentence is being referred to in "Lines 200-201"?

Thanks for your time and effort in writing your useful review and I look forward to hearing from you.

Nick

---

## Editor Comment (EC1) · B. Schaefli (Editor) · 27 Nov 2017

For some technical reasons (problem in the system?), the reviewer received an earlier version of the manuscript for review, which should be content-wise almost the same as the the one in the public discussion but which did not have the same line numbers. Here the sentences to which the mentioned line numbers correspond to:

16 - 20: In rivers with limited variation in the physical properties of the suspended sediment, an increase in suspended-sediment concentration will initially cause a linear increase in turbidity. When the suspended-sediment concentration in these rivers causes turbidity levels that exceed the upper measurement limit of a probe, turbidity

probes do not necessarily "peg" at a constant value.

32-33: Although turbidity is commonly used to monitor change in water clarity, it is not an absolute measure of water clarity in part because it is an instrument-specific measurement.

104: Physical suspended-sediment samples are collected episodically to verify the 104 acoustical and turbidity measurements.

174-175: The sediment concentration at which false low turbidity occurs will depend on the characteristics of the sediment, particularly the grain-size distribution, as well as the properties of the instrument, such as path length and detector angle.

200-201: One indicator an instrument is recording false low turbidity could be a pattern of lower values bracketed by values at the maximum recording level. Data showing this pattern should be verified using surrogate measures of turbidity, such as acoustic attenuation or suspended-sediment concentration, especially if suspended-sediment concentration is known or suspected to be particularly high (e.g. several thousand 203 mg/L)

---

## Referee Comment (RC2) · Anonymous Referee #2 · 28 Nov 2017

This paper deals with false low turbidity readings, using an optical probe, which occur at a turbidity level well above the nominal range of the probe. It is a very specific issue and it is the first time I read about these false readings. The subject of this study is relevant with themes of the HESS journal. The findings are based on field measurements on the Colorado River and lab experiments. I suggest accepting this technical note for publication after the following clarifications.

In my opinion, few points should be clarify in the abstract (and more generally in the paper): Are the conclusions valid for all type of optical probes and with different sorts of suspended matter? Because for instance nephelometers are affected by grain size.

[Figure]

Should we expect the same conclusions with nephelometer and turbidimeter, or between forward-scattering, side-scattering and back-scattering (in the case of Nephelometry)?

Concerning the purposes of monitoring turbidity, I would add two other points: 1) "to study sediment transport in river and catchment erosion", and 2) "to manage water reservoir (silting and emptying)". Some references relative to the five purposes may be added (possibly in the introduction section).

I think it could be useful to give some tips for detecting these false low turbidity readings. If another surrogate measurement of turbidity is not available, could these low false turbidity measurements be detected? Note that values of silt and clay concentration to obtain false low turbidity readings are around ten times higher than maximum concentration measurable by the instrument tested in this study. This paper could also be useful to underline the importance of the instrument choice, notably about the nominal range of the probe.

In the second paragraph of the introduction, the authors should add a definition of nephelometric (determination of light-scattering).

In the third paragraph of the introduction, the authors should mention the type of data that is used in figure 2 and in ASTM International (2011) to deduce the different responses. A reference to figure 5 should be given in line 26 of the introduction to provide an example (or a citation should be added).

In section 2.2, the authors should present briefly the experimental setup: equipment used with dimensions (in particular relative to the conditions required to use properly the probe).

In section 3.1, I suggest to use the station names presented in figure 3 (in the second sentence). In the third sentence, "at or above" is confusing to me. I understood that false low turbidity values are observed only above probe maximum recording level

(never "at" maximum).

Considering section 3.2, apparently only one experiment has been conducted. Few words about the uncertainties and/or the reproducibility could be welcome (in relation with the two different probes). The threshold value of 38,000 mg/L observed in the experiment should be also compared with the range 17,000 mg/L to 27,000 mg/L from the field.

In section 4, as "false low turbidity occurred during approximately 70 percent of the suspended-sediment load of the flooding event", it should be mentioned that for the Colorado River it seems to be necessary to use another probe with higher saturation level.

In figure 4, few axis ticks are missing.

In figure 5, for clarity, I suggest to extend the graph to 30th of September to see the end of the plateau and the decreasing turbidity following the event.

Finally, I noticed that, although the authors used the same model of instrument for lab and field measurement, the saturation occurred at 1700 FNU and around 1500 FNU respectively. However, it might not have an influence on the conclusions of the study.

Here, I cite two papers that might be helpful to understand the complexity of turbidity measurement: Kitchener, B. G., Wainwright, J., & Parsons, A. J. (2017). A review of the principles of turbidity measurement. Progress in Physical Geography, 41(5), 620-642. Ziegler, A. C. (2002, April). Issues related to use of turbidity measurements as a surrogate for suspended sediment. In Turbidity and other sediment surrogates workshop (Vol. 1).

---

## Author Response (AR1)

[revised manuscript text omitted]

**Relevant Changes to the Manuscript**

1. The word 'optical' was added to the title and abstract.
2. Abstract
   a. Two additional purposes for monitoring turbidity were added.
   b. Reworded to clarify the change in recorded turbidity at different suspended-sediment concentrations and the recording of false low turbidity.
3. 1 Introduction
   a. Paragraph 1
      i. Reworded to clarify why turbidity is not an absolute measure of water clarity.
   b. Paragraph 2
      i. Added definition of nephelometric turbidity probe.
      ii. Added wavelength of a typical turbidity probe (used in our study).
   c. Paragraph 3
      i. Added details describing Figure 2 and the different turbidity responses to varying concentration. Decided not to describe in detail the horizontal axis of Figure 2 in the text but did describe it in the figure caption.
      ii. Added a reference to Figure 5.
4. 2 Methods
   a. 2.1 Field sites
      i. Added the type of acoustic instruments and the methods used for collecting suspended-sediment samples. Also added a reference for the suspended-sediment sampling.
   b. 2.2 Laboratory experiment
      i. Added the type and a description of the electric stirrer, the size of the bucket, the placement of the stirrer and turbidity probe in the bucket, the type of water quality instrument used, and details on the use of the electric stirrer.
5. 3 Results
   a. 3.1 Field data
      i. Added names of stations.
      ii. The words "at or" were removed from "at or above".
      iii. Added phrase to clarify that false low turbidity readings are recorded at high but variable silt and clay concentrations.
6. 4 Discussion and conclusions
   a. Paragraph 2
      i. Added sentence describing the main sediment characteristics affecting turbidity readings.
      ii. Added comparison of lab and field results and the likely reason for their difference. Note that after discussions with co-authors, we decided to only add the most likely reason for the difference between lab and field results rather than list other potential reasons (such as flocculation) which are much less likely.
   b. Paragraph 3
      i. This paragraph was added to
         1. Discuss how different turbidity probe characteristics may affect the recording of false low turbidity.
         2. Introduce multiple detector probes as a possible solution for avoiding false low turbidity.

> > > 3. Briefly discuss that multiple detector probes may not always be a practical choice and that the choice of a turbidity probe may be limited by the choice of a water quality instrument.
> > c. Last paragraph
> > > i. Added a sentence describing why the mentioned turbidity pattern would be expected during a period when false low turbidity was recorded.
> 7. References
> > a. Added reference (Edwards and others, 1999) describing suspended sediment collection techniques (referred to in section 2.1).
> 8. Figure 2
> > a. Added a detailed description of the horizontal axis.
> 9. Figure 4
> > a. Added a legend (Explanation).
> 10. Figure 5
> > a. Extended the date range to include turbidity and silt and clay measurements following the flooding event.
> > b. Added details in figure caption of when false low turbidity was recorded.

**Response to Referee #1 comments:**

Thank you for the comments and suggestions. They will result in an improved the paper. Below are my responses to the all of the comments from Referee #1.

The word 'optical' will be put in the title.

The word 'optical' will also be added in the abstract to stress that the issue pertains to optical turbidity probes.

10 The sentences in lines 16-20 in the abstract will be changed to clarify the response from optical turbidity probes seen at low and high suspended-sediment concentrations, and when false low turbidity readings may be seen.

In lines 32-33, the sentence will be clarified to explain why turbidity is not an absolute measure of water clarity.

15 In section 2.1 (line 104 in Referee #1 comments), the type of acoustical instrument and the methods used for collecting the suspended-sediment samples will be added. A reference for the suspended-sediment sample collection methods will also be added. Because this is a short technical note, no other details pertaining to the acoustical data collection or suspended-sediment sampling will be included, although a thorough description can be obtained from the listed references.

20 A legend will be included with Figure 4.

In the 2nd paragraph of section 4 (lines 174-175 in Referee #1 comments), a more thorough discussion will be presented on how sediment characteristics and instrument properties affect turbidity.

25 In the final paragraph of section 4 (lines 200-201 in Referee #1 comments), a sentence will be added explaining why false low turbidity would likely show a pattern of turbidity within the valid measurement range of the probe bracketed by pegged turbidity.

**Response to Referee #2 comments:**

The comments and suggestions made by Referee #2 are useful and will result in an improved paper.

5   I will address how false low turbidity relates to grain size and whether it is likely to occur with different types of turbidity instruments. This will be addressed in section 4.

I will include in the abstract the two additional suggestions for purposes for monitoring turbidity.

10   The last paragraph in the paper addresses the expected turbidity pattern when false low turbidity is present. I will make this more clear (as stated in my response to Referee #1 comments). I will also talk about instrument choice and expand on the discussion of the range of turbidity values when false low turbidity was present in the lab and field.

I will add a definition of a nephelometric probe in the 2nd paragraph of the introduction.

Additional details will be included pertaining to figure 2 in the 3rd paragraph of the introduction. A reference to figure 5 will be given in the same paragraph after referring to false low turbidity at the Grand Canyon site.

Additional details will be included in section 2.2 to describe the experimental setup with the model of electric stirrer mentioned,
20   the size of bucket used, and the model of instrument used (the model of turbidity probe is already mentioned). I will also mention the distance of the stirrer and turbidity probe from the bottom of the bucket.

In section 3.1, station names will be added and "at" will be removed from "at or above" in the 3rd sentence.

25   In response to comments regarding section 3.2, the 38,000 mg/L lab value will be compared to the 17,000 to 27,000 mg/L range observed in the field in the 2nd paragraph of section 4. I will mention that there are uncertainties with the lab experiment and I will list some reasons that may help explain the significant difference between the lab experiment value and the field values: a single sediment source from the field site was used for the lab experiment with unknown grain size (in the silt and clay size range) whereas the field readings are the result of sediment from many sources different than the sediment used for
30   the lab experiment; the physical behavior of the silt and clay in the lab experiment (i.e. flocculation) is likely different than what occurs in the field, resulting in different turbidity readings.

Referee #2 noted that it should be mentioned that "it seems necessary to use another probe with a higher saturation level" in our study area. There are several reasons we use the particular probe that we do: it is the only choice we have with the water quality monitor that we use (which we also use to record temperature, specific conductance, dissolved oxygen, and pH); it is capable of accurately detecting lower turbidity levels which are important to characterize biological processes; we have other

5   surrogate measures of suspended sediment (acoustic) which accurately record at our highest sediment concentrations. I believe that further explaining our choice of turbidity probe is beyond the scope of this short technical note which focuses on false low turbidity. However, when I talk about how false low turbidity relates to different types of turbidity sensors in section 4, I will mention some considerations for choosing turbidity probes.

10   The missing tick marks in figure 4 are actually not missing in the .pdf version of the figure.

I will extend the date range in figure 5 to show turbidity following the flooding event.

Referring to turbidity saturation being different in the lab and field experiment (approx. 1700 FNU vs. 1500 FNU, shown in

15   figures 4 and 5), each YSI 6136 turbidity probe has a slightly different maximum recording level. I don't plan on mentioning this since it is not significant when considering false low turbidity.